

# Icariin affects cell cycle progression and proliferation of human retinal pigment epithelial cells via enhancing expression of H19

Yibing Zhang[1], Min Li[2] and Xue Han[1]

[1] Department of Ophthalmology, The First Hospital of Jilin University, Changchun, China
[2] Department of Pharmacology and Toxicology, Jilin University School of Pharmaceutical Sciences, Changchun, China

## ABSTRACT

**Background:** Aberrant proliferation of retinal pigment epithelial (RPE) cells under pathologic condition results in the occurrence of proliferative vitreoretinopathy (PVR). Icariin (ICA)-a flavonol glucoside-has been shown to inhibit proliferation of many cell types, but the effect on RPE cells is unknown. This study aimed to clarify the inhibitory effects of ICA on RPE cells against platelet-derived growth factor (PDGF)-BB-induced cell proliferation, and discuss the regulatory function of H19 in RPE cells.

**Methods:** MTS assay was conducted to determine the effects of ICA on cell proliferation. Flow cytometry analysis was performed to detect cell cycle progression. Quantitative real-time PCR and western blot assay were used to measure the expression patterns of genes in RPE cells.

**Results:** ICA significantly suppressed PDGF-BB-stimulated RPE cell proliferation in a concentration-dependent manner. Moreover, since administration of ICA induced cell cycle G0/G1 phase arrest, the anti-proliferative activity of ICA may be due to G0/G1 phase arrest in RPE cells. At molecular levels, cell cycle regulators cyclin D1, CDK4, CDK6, p21 and p53 were modulated in response to treatment with ICA. Most importantly, H19 was positively regulated by ICA and H19 depletion could reverse the inhibitory effects of ICA on cell cycle progression and proliferation in PDGF-BB-stimulated RPE cells. Further mechanical explorations showed that H19 knockdown resulted in alternative expressions levels of cyclin D1, CDK4, CDK6, p21 and p53 under ICA treatment.

**Conclusions:** Our findings revealed that ICA was an effective inhibitor of PDGF-BB-induced RPE cell proliferation through affecting the expression levels of cell cycle-associated factors, and highlighted the potential application of ICA in PVR therapy. H19 was described as a target regulatory gene of ICA whose disruption may contribute to excessive proliferation of RPE cells, suggesting that modulation of H19 expression may be a novel therapeutic approach to treat PVR.

Corresponding author
Yibing Zhang,
zhangyibing220@jlu.edu.cn

## INTRODUCTION

Proliferative vitreoretinopathy (PVR), a defective wound repair process responding to either retinal detachment or vitreoretinal surgery, is a severe blinding disease due to the recurrent epiretinal membranes (ERMs) traction (*Feng et al., 2013*). PVR is characterized by proliferation of cells and contraction of membranes within the vitreous cavity and on both sides of the retinal surfaces, which is a common cause of failure in rhegmatogenous retinal detachment surgery (*Khan, Brady & Kaiser, 2015*). Many attempts have been made in PVR pharmacotherapy (*Abdullatif et al., 2018*; *Tikhonovich, Erdiakov & Gavrilova, 2018*), but there are not ideal drugs for PVR treatment because the molecular mechanisms underlying PVR are not well understood. Pathogenesis of PVR is a complex biological process regulated by multiple growth factors and cytokines (*Hoerster et al., 2017*), and the retinal pigment epithelium (RPE) cells are considered as the key cell type in PVR, which is deemed to proliferate and migrate through retinal breaks (*Umazume et al., 2014*; *Chen et al., 2013*). Among the growth factors, platelet-derived growth factor (PDGF)-induced RPE cell proliferation is an important step for promoting fibrotic membrane formation in PVR development (*Li et al., 2007*). There are four PDGF isoforms (PDGF-A, -B, -C and -D) that form homodimers or heterodimers (PDGF-AA, -BB, AB, -CC and -DD) through disulfide bonds. Moreover, PDGF-BB shows extensively proliferative effects on RPE cells (*Chan et al., 2009*, *2013*; *Zhang et al., 2019*; *Li et al., 2018b*). Since the proliferation of activated RPE cells is believed to be a central event in the pathogenesis of PVR, it is vital to identify specific drug targets whose modulation can block the uncontrolled proliferation of RPE cells.

Icariin (ICA) (molecular formula: $C_{33}H_{40}O_{15}$; molecular weight: 676.67 g/mol), the main active ingredient of the traditional Chinese medical plant *Herba Epimedii*, has a wide range of pharmacological and biological activities, such as possessing anti-oxidative, anti-tumor and anti-inflammatory activities (*Li et al., 2015*; *Zhang et al., 2015*; *Fang & Zhang, 2017*). Recently, emerging studies have reported ICA regulates cell cycle progression and proliferation in various cell types. ICA exerted suppressive effects on medulloblastoma cells (*Sun et al., 2016b*), osteosarcoma cells (*Ren, Zhu & Liu, 2018*) and human aortic smooth muscle cells (HA-VSMCs) (*Hu et al., 2016*). Further, the induction of G0/G1 phase arrest was observed in cardiomyocyte and breast cancer cells after treatment with ICA (*Zhu et al., 2005*; *Cheng et al., 2019*). However, whether ICA could modulate cell cycle progression and proliferation in RPE cells remains largely unclear.

Long noncoding RNAs (lncRNAs) are transcripts with a length >200 nucleotides structurally homologous to coding mRNAs, but has little or no protein-coding potential (*Ma, Bajic & Zhang, 2013*). LncRNAs can modulate gene expression and are implicated in diverse biological processes, such as cell differentiation, cell cycle, cell proliferation, apoptosis, migration and stem cell maintenance (*Mercer & Mattick, 2013*). Dysregulation of lncRNAs is found involved in multiple human diseases such as cancer, cardiovascular diseases, and neurological problems (*Shi et al., 2013*; *Zhang et al., 2013*; *Sun et al., 2016a*; *Li et al., 2018c*; *Kour & Rath, 2017*). Roles of lncRNAs in ocular disorders, such as ocular tumors, diabetic retinopathy (DR), PVR, cataract and glaucoma, have also been

identified (*Zhou et al., 2015*). Moreover, as one of the best-characterized lncRNAs with multiple functions, H19 has been closely associated with the pathogenesis of ocular diseases in recent studies (*Klein et al., 2016*; *Thomas et al., 2019*; *Liu et al., 2018*). Nevertheless, the effect of H19 on cell cycle and proliferation in RPE cells is still uninvestigated.

In this study, we found that ICA treatment could significantly cause delays at the G1-S transition and inhibit RPE cell proliferation upon the stimulus of PDGF-BB, which may be realized by affecting the expression of cell cycle-related genes, such as cyclin D1, cyclin-dependent kinase 4 (CDK4), CDK6, CDK inhibitor 1A (p21) and p53. We further explored the biological function of H19 which might contribute to the inhibitory effect of ICA, describing an experimental basis for investigating H19 as a potential therapeutic target for PVR. Our findings highlighted the anti-proliferative effects of ICA, as well as determined the potential influences of H19 in RPE cells, providing a new perspective for the treatment of PVR.

## MATERIALS AND METHODS

### Cell culture

ARPE-19 cells were purchased from American Type Culture Collection (ATCC, Manassas, VA, USA) and maintained in DMEM/F12 medium (Invitrogen, Carlsbad, CA, USA) supplemented with 10% FBS, 100 U/ml penicillin and 100 mg/ml streptomycin (all from Invitrogen) at 37 °C in a humidified atmosphere with 5% $CO_2$.

### Cell treatment and transfection

ICA (Vic's Biological Technology Co., Ltd., Sichuan, China) was dissolved in DMSO (Sigma–Aldrich, St. Louis, MO, USA) and stocked at 100 mM, which was diluted to the desired concentrations with culture medium. When tested, ICA was added 30 min prior to PDGF-BB (20 ng/ml; Selleck Chemicals, Houston, TX, USA) stimulation.

Small interfering RNA oligonucleotides targeting H19 (si-H19) was obtained from Ribobio Biotechnology (Guangzhou Ribobio Co., Ltd., Guangzhou, China). The scrambled siRNA (si-NC) was used as a negative control. RPE cells were transfected with si-H19 or si-NC by using Lipofectamine 2000 reagent (Invitrogen, Carlsbad, CA, USA) following the manufacturer's protocol. After transfected for 48 h, cells were harvested for the following experiment and divided into five groups: (1) blank control group without PDGF-BB; (2) PDGF-BB-stimulated group (with PDGF-BB 20 ng/ml only); (3) PDGF-BB +ICA treatment group (with PDGF-BB and ICA); (4) PDGF-BB+ICA+si-NC treatment group (with PDGF-BB, ICA and si-NC); (5) PDGF-BB+ICA+si-H19 treatment group (with PDGF-BB, ICA and si-H19).

### MTS assay

Retinal pigment epithelium cells were cultured in 96-well plate at a density of $1 \times 10^4$ cells/well, and incubated for 24 h under 5% $CO_2$ at 37 °C. After different treatments, these cells were collected at each time point (0, 24, 48 or 72 h). Cell proliferation was then assessed by the colorimetric MTS (3-(4,5-dimethylthiazol-2-yl)-5-(3-carboxymethoxyphenyl)-2-(4-sulfophenyl)-2H-tetrazolium) reduction method
**Table 1 Primers used for qPCR.**

| Gene | Forward primer (5′–3′) | Reverse primer (5′–3′) |
|------|------------------------|------------------------|
| p53 | TAGTGTGGTGGTGCCCTATG | CCAGTGTGATGATGGTGAGG |
| p21 | TGCCCAAGCTCTACCTTCC | CAGGTCCACATGGTCTTCCT |
| CDK4 | ATGTCCGACCTGTTCCACA | CGAAGTCAAAGTTCCACCG |
| CDK6 | ACGTGGTCAGGTTGTTT | TTTATGGTTTCAGTGGG |
| cyclin D1 | CCCGCACGATTTCATTGAAC | AGGGCGGATTGGAAATGAAC |
| H19 | GCGGGTCTGTTTCTTTACTTC | GTGGTTGTAAAGTGCAGCAT |

(Cell Titer 96® Aqueous One Solution Reagent; Promega, Madison, WI, USA) following the manufacturer's instructions. The absorbance was measured at 490 nm with a microplate reader (Thermo Fisher, Waltham, MA, USA).

## Flow cytometry analysis

The cell cycle distribution of RPE cells was estimated using a flow cytometer by quantitation of DNA content of cells stained with propidium iodide (PI). In brief, RPE cells were seeded on 6-cm dishes and harvested after 48 h. Then cells were fixed overnight at 4 °C with 70% ethanol, followed by resuspension in 500 μl of PBS. Fixed cells were incubated in PBS containing PI and RNase A for at least 30 min at 4 °C. Cellular DNA content was analyzed on a BD FACSCalibur (Becton Dickinson, San Jose, CA, USA). Data was analyzed with ModFit LT 3.0 software (Variety Software House, Inc., Topsham, ME, USA).

## Quantitative real-time PCR

Total RNA was isolated from RPE cells using TRIzol reagent (Invitrogen, Carlsbad, CA, USA) and transcribed into cDNA using an ImProm-IITM Reverse Transcription System (Promega, Madison, WI, USA) according to the protocol provided by the manufacturer. Quantitaitve PCR was performed using SYBR GREEN Quantitative Real-time PCR (qPCR) Super Mix (Invitrogen, Carlsbad, CA, USA) in a 7500 Real-Time PCR system (Applied Biosystems, Carlsbad, CA, USA). The $2^{-\Delta\Delta Ct}$ method was used to quantify the relative mRNA expression changes (*Livak & Schmittgen, 2001*), using GAPDH as an internal control. All forward and reverse primers were synthesized by Sangon Biotech (Sangon Biotech Co., Ltd., Shanghai, China) and are indicated in Table 1.

## Western blot analysis

Retinal pigment epithelium cells were collected and lysed in radio-immunoprecipitation buffer (Beyotime Biotechnology Co., Ltd., Shanghai, China), and then centrifuged at 14,000 rpm for 10 min at 4 °C. The protein concentration of the supernatants was determined using the BCA protein assay (KeyGEN Biotech Co., Ltd., Nanjing, China). Total proteins were separated by electrophoresis on 10% SDS-polyacrylamide gels, and the proteins were electroblotted onto polyvinylidene fluoride membranes (Millipore, Billerica, MA, USA). After blocking with 5% skimmed milk for 1 h at room temperature, the membranes were incubated with primary antibodies against p53 (diluted at 1:1,000,

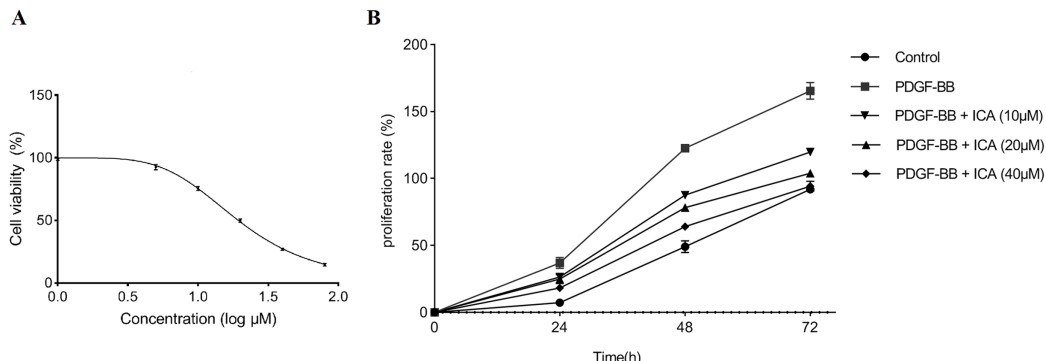

**Figure 1 Cell viability was assessed in RPE cells by using the MTS assay.** (A) ICA treatments exerted an inhibitory effect on RPE cells without stimulation of PDGF-BB. After a series concentration of ICA (0, 1, 5, 10, 20, 40 and 80 μM) treatments for 48 h, the IC50 value of ICA was calculated using GraphPad Prism 8.0. (B) The cellular proliferating ability was assessed by the proliferation curve at 0, 24, 48 and 72 h after ICA treatment. Control: blank control group without PDGF-BB; PDGF-BB: 20 ng/ml PDGF-BB; PDGF-BB + ICA (10 μM): 20 ng/ml PDGF-BB + 10 μM ICA; PDGF-BB + ICA (20 μM): 20 ng/ml PDGF-BB + 20 μM ICA; PDGF-BB + ICA (40 μM): 20 ng/ml PDGF-BB + 40 μM ICA. Data are expressed as mean ± SD.

ab131442), p21 (diluted at 1:1,000, ab109520), CDK4 (diluted at 1:1,000, ab108357), CDK6 (diluted at 1:1,000, ab124821), cyclinD1 (diluted at 1:1,000, ab134175), and GAPDH (1:1,000, ab181602; all from Abcam, San Diego, MA, USA) at 4 °C overnight. After being incubated with horseradish peroxidase-conjugated secondary antibodies for 1 h at 37 °C, the membranes were developed with chemiluminescence substrate (Millipore, Billerica, MA, USA) and quantified with Image J system (BioRad, Hercules, CA, USA).

## Statistical analysis

All quantitative data were presented as mean ± standard deviation (SD). At least three independent experiments were done. All statistical parameters were calculated with GraphPad Prism 8.0 (GraphPad Software Inc., San Diego, CA, USA). Student's *t* test was used to analyze the difference between two groups. One-way ANOVA followed by post-hoc test with least significant difference was performed to evaluate differences among multiple groups. $P < 0.05$ was considered statistically significant.

## RESULTS

### ICA decreased viability of RPE cells in a concentration-dependent manner

The inhibitory effect of ICA on the RPE cells without stimulation of PDGF-BB was detected via MTS assay initially. ICA concentrations were set as 1, 5, 10, 20, 40 and 80 μM, and the blank control were established. Compared with control group, we found that ICA treatment significantly decreased the viability radio of RPE cells in a concentration-dependent manner, the half maximal inhibitory concentration (IC50) value of ICA was 19.36 μM (Fig. 1A).

**Table 2 Inhibitory effects of ICA with different concentrations on PDGF-BB-stimulated proliferation of RPE cells.**

| Group | 24 h | | 48 h | | 72 h | |
|---|---|---|---|---|---|---|
| | Absorbance | Inhibition (%) | Absorbance | Inhibition (%) | Absorbance | Inhibition (%) |
| Control | $0.38 \pm 0.01$ | | $0.53 \pm 0.02$ | | $0.69 \pm 0.01$ | |
| PDGF-BB | $0.48 \pm 0.01^{\#\#\#}$ | | $0.78 \pm 0.01^{\#\#\#}$ | | $0.94 \pm 0.02^{\#\#\#}$ | |
| PDGF-BB + ICA (10 µM) | $0.45 \pm 0.01^{*}$ | 28.53 | $0.67 \pm 0.01^{***}$ | 43.88 | $0.77 \pm 0.01^{***}$ | 64.26 |
| PDGF-BB + ICA (20 µM) | $0.44 \pm 0.01^{*}$ | 40.28 | $0.63 \pm 0.01^{***\Delta\Delta}$ | 60.62 | $0.72 \pm 0.01^{***\Delta\Delta\Delta}$ | 86.40 |
| PDGF-BB + ICA (40 µM) | $0.43 \pm 0.01^{**\Delta}$ | 55.93 | $0.59 \pm 0.01^{***\Delta\Delta\Delta\Diamond\Diamond}$ | 76.37 | $0.68 \pm 0.01^{***\Delta\Delta\Delta\Diamond}$ | 99.97 |

Notes:
* $P < 0.05$ vs. PDGF-BB group.
** $P < 0.01$ vs. PDGF-BB group.
*** $P < 0.001$ vs. PDGF-BB group.
### $P < 0.001$ vs. control group.
$\Delta$ $P < 0.05$ vs. 20 ng/ml PDGF-BB + 10 µM ICA group.
$\Delta\Delta$ $P < 0.01$ vs. 20 ng/ml PDGF-BB + 10 µM ICA group.
$\Delta\Delta\Delta$ $P < 0.001$ vs. 20 ng/ml PDGF-BB + 10 µM ICA group.
$\Diamond$ $P < 0.05$ vs. 20 ng/ml PDGF-BB + 20 µM ICA group.
$\Diamond\Diamond$ $P < 0.01$ vs. 20 ng/ml PDGF-BB + 20 µM ICA group.
RPE cells proliferation was determined with MTS assay at 24, 48 and 72 h after ICA treatment. Absorbance at 490 nm shows that ICA obviously inhibited PDGF-BB-stimulated proliferation of RPE cells in a concentration dependent manner. The IC50 values of ICA in RPE cells were 30.13 µM at 24 h and 12.53 µM at 48 h. Control: blank control group without PDGF-BB; PDGF-BB: 20 ng/ml PDGF-BB; PDGF-BB + ICA (10 µM): 20 ng/ml PDGF-BB + 10 µM ICA; PDGF-BB + ICA (20 µM): 20 ng/ml PDGF-BB + 20 µM ICA; PDGF-BB + ICA (40 µM): 20 ng/ml PDGF-BB + 40 µM ICA. Data are expressed as mean ± SD.

Moreover, the effect of ICA on PDGF-BB-stimulated proliferation of RPE cells was tested with increasing concentration of ICA (10–40 µM) and the rate of proliferation was calculated. The results were shown in Table 2 and Fig. 1B. As compared with blank control group, the absorbance in RPE cells was significantly enhanced by PDGF-BB stimulation for 24–72 h; contrarily, we observed that ICA showed concentration-dependent anti-proliferative activity in RPE cells.

## ICA induced cell cycle G0/G1 phase arrest in PDGF-BB-stimulated RPE cells

We used flow cytometer analysis to test the effect of ICA on cell cycle progression. ICA-induced inhibition of RPE cell proliferation was associated with G0/G1 arrest. The proportion of RPE cells in the G0/G1 phase was significantly lower in the PDGF-BB-stimulated group than in the control group. After the RPE cells were incubated with ICA (10, 20 and 40 µM) for 48 h, the total percentage of RPE cells in G0/G1 phase increased, significantly when concentration of ICA reached 40 µM ($P < 0.001$ vs. PDGF-BB group), suggesting that ICA delayed the G1/S phase transition (Fig. 2).

## ICA affected the expression levels of cell cycle regulators in RPE cells

Results in Figs. 1 and 2 displayed that ICA remarkably suppressed cell cycle progression and proliferation of RPE cells when its concentrations was 40 µM. Therefore, 40 µM ICA was used to treat RPE cells in the following experiment. The expression levels of cell cycle regulators (CDK4, CDK6, cyclin D1, p21 and p53) were further evaluated by qPCR and western blot. After 48 h of stimulation with PDGF-BB, it was found that the mRNA levels of p53 and p21 were decreased, while showed increase in CDK4, CDK6 and cyclin D1. Consistently with G0/G1 arrest, ICA increased the mRNA levels of p53 and p21,

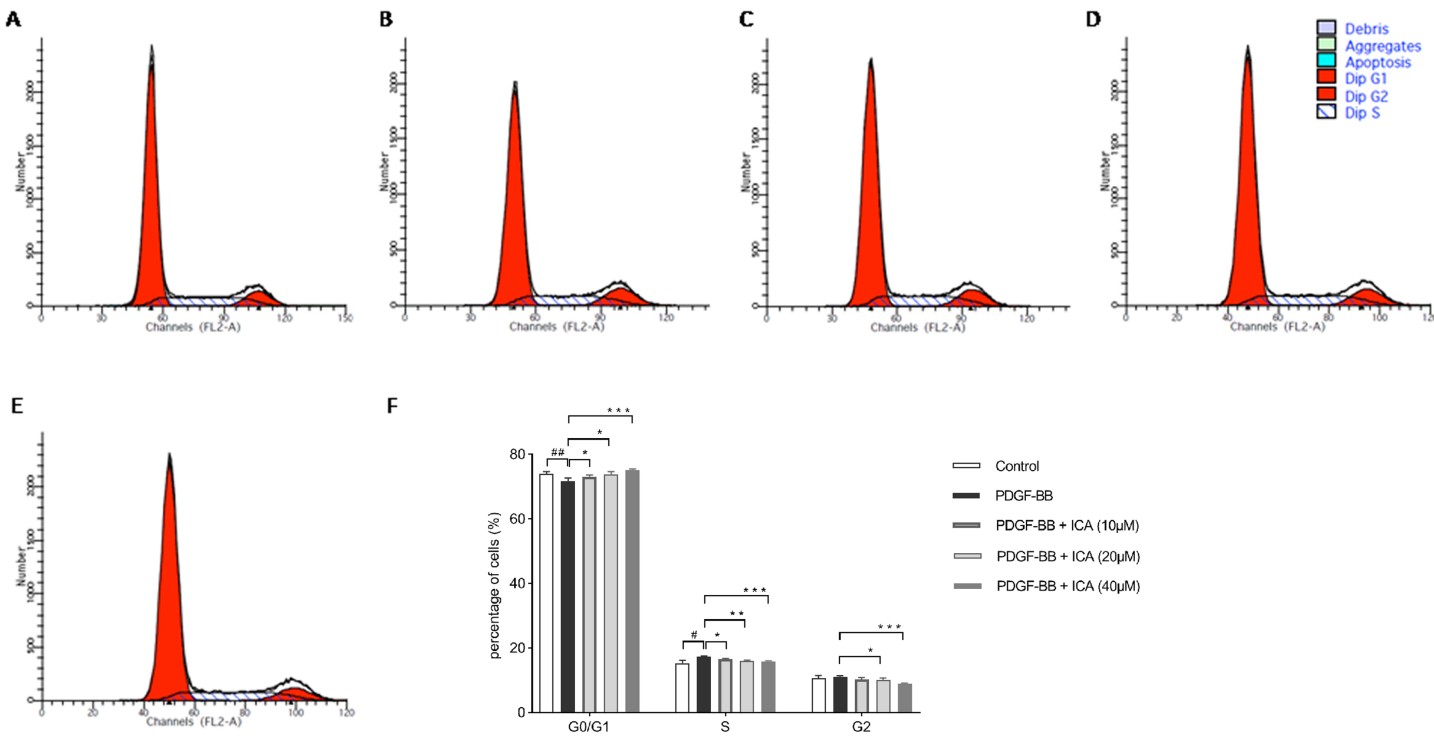

**Figure 2 Effects of ICA on the cell cycle progression of PDGF-BB-stimulated RPE cells after 48 h.** Cell cycle was determined by individual nuclear DNA content reflected by fluorescence intensity of incorporated PI. Percentage of cells in G0/G1, S and G2/M phases were calculated. Represent images of cell cycle in (A) Control; (B) PDGF-BB; (C) PDGF-BB + ICA (10 μM); (D) PDGF-BB + ICA (20 μM) and (E) PDGF-BB + ICA (40 μM); (F) bar graphs show the percentage of cells at each stage. Data are expressed as mean ± SD ($n = 3$). $^{\#}P < 0.05$, $^{\#\#}P < 0.01$ vs. control group; $^{*}P < 0.05$, $^{**}P < 0.01$, $^{***}P < 0.001$ vs. PDGF-BB group.

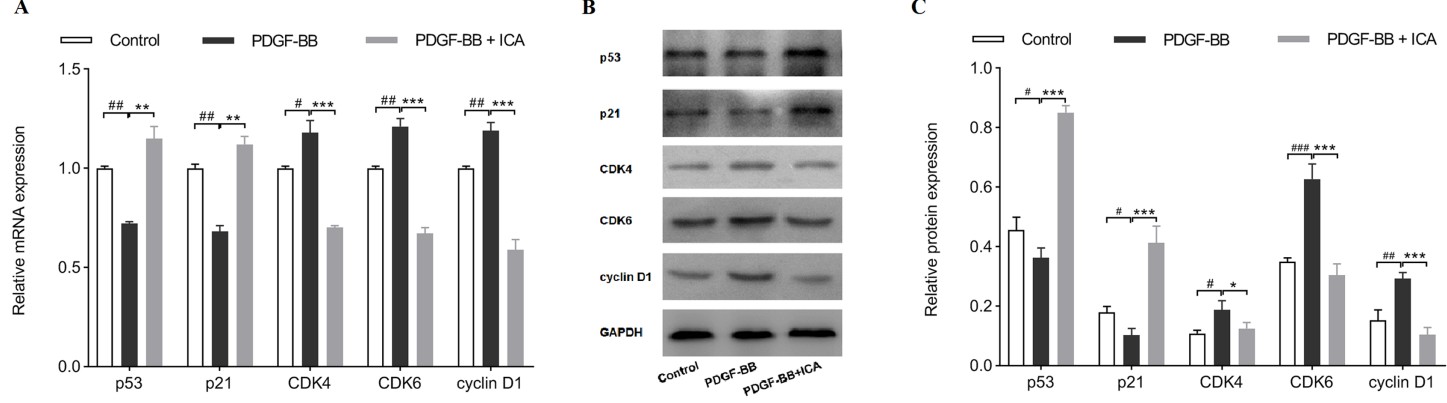

**Figure 3 Effects of ICA on cell cycle-related factors expression in PDGF-BB-stimulated RPE cells.** The mRNA levels of p53, p21, CDK4, CDK6 and cyclin D1 were determined by qPCR (A), and the protein levels of p53, p21, CDK4, CDK6 and cyclin D1 were detected by western blot assay (B and C). Control: blank control group without PDGF-BB; PDGF-BB: 20 ng/ml PDGF-BB; PDGF-BB + ICA: 20 ng/ml PDGF-BB + 40 μM ICA. Data are expressed as mean ± SD ($n = 3$). $^{\#}P < 0.05$, $^{\#\#}P < 0.01$, $^{\#\#\#}P < 0.001$ vs. control group; $^{*}P < 0.05$, $^{**}P < 0.01$, $^{***}P < 0.001$ vs. PDGF-BB group.

while reduced the mRNA levels of CDK4, CDK6 and cyclin D1 (Fig. 3A). At protein levels, p53 and p21 were also decreased after PDGF-BB stimulation, while CDK4, CDK6 and cyclin D1 were increased (Figs. 3B and 3C). Comparatively, the protein levels of p53 and

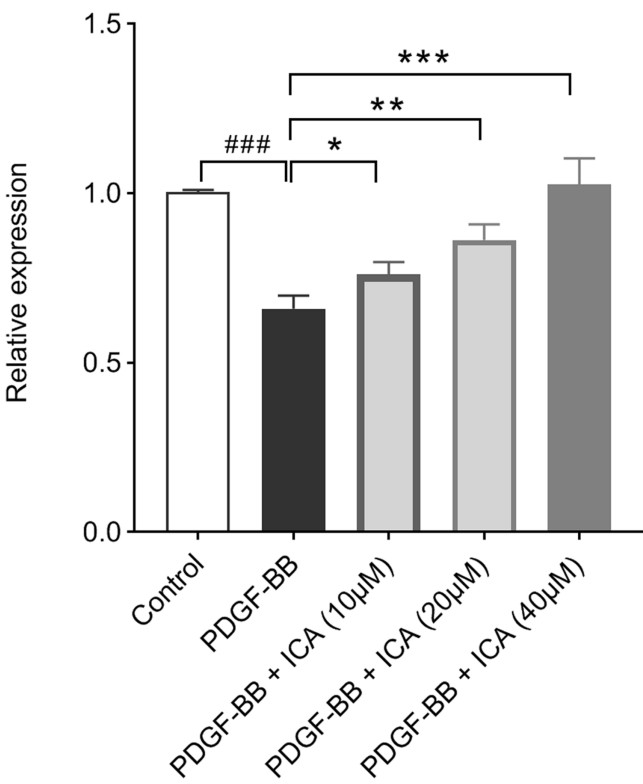

**Figure 4 Effects of ICA on PDGF-BB-stimulated H19 expression in RPE cells.** The expression of H19 as determined by qPCR was decreased by PDGF-BB, but ICA increased H19 expression depending upon concentrations. Data are expressed as mean ± SD. [###]$P < 0.001$ vs. control group; [*]$P < 0.05$, [**]$P < 0.01$, [***]$P < 0.001$ vs. PDGF-BB group.               

p21 were increased in response to treatment with ICA in PDGF-BB-stimulated RPE cell. Likewise, CDK4, CDK6 and cyclin D1 were significantly reduced at protein levels.

## ICA enhanced the expression level of H19 in PDGF-BB-stimulated RPE cells

The RPE cells were treated with different concentrations of ICA and then exposed to PDGF-BB stimulation, after which qPCR was undertaken. The results showed that H19 expression level was significantly down-regulated by PDGF-BB, ICA could increase the expression level of H19 in a concentration-dependent manner (Fig. 4).

## ICA inhibited the PDGF-BB-stimulated cell proliferation via enhancing the expression of H19

Afterward, si-H19 and si-NC were transfected into RPE cells, and the transfected efficiency was presented in Fig. 5A. The expression level of H19 was markedly down-regulated in si-H19-transfected cells compared with that in si-NC-transfected cells. The data indicated that si-H19 was successfully transfected into RPE cells to inhibit H19 expression.

Down-regulation of H19 could partially reverse the inhibition of cell proliferation induced by ICA treatment in RPE cells (Figs. 5B and 5C). Subsequently, the increased

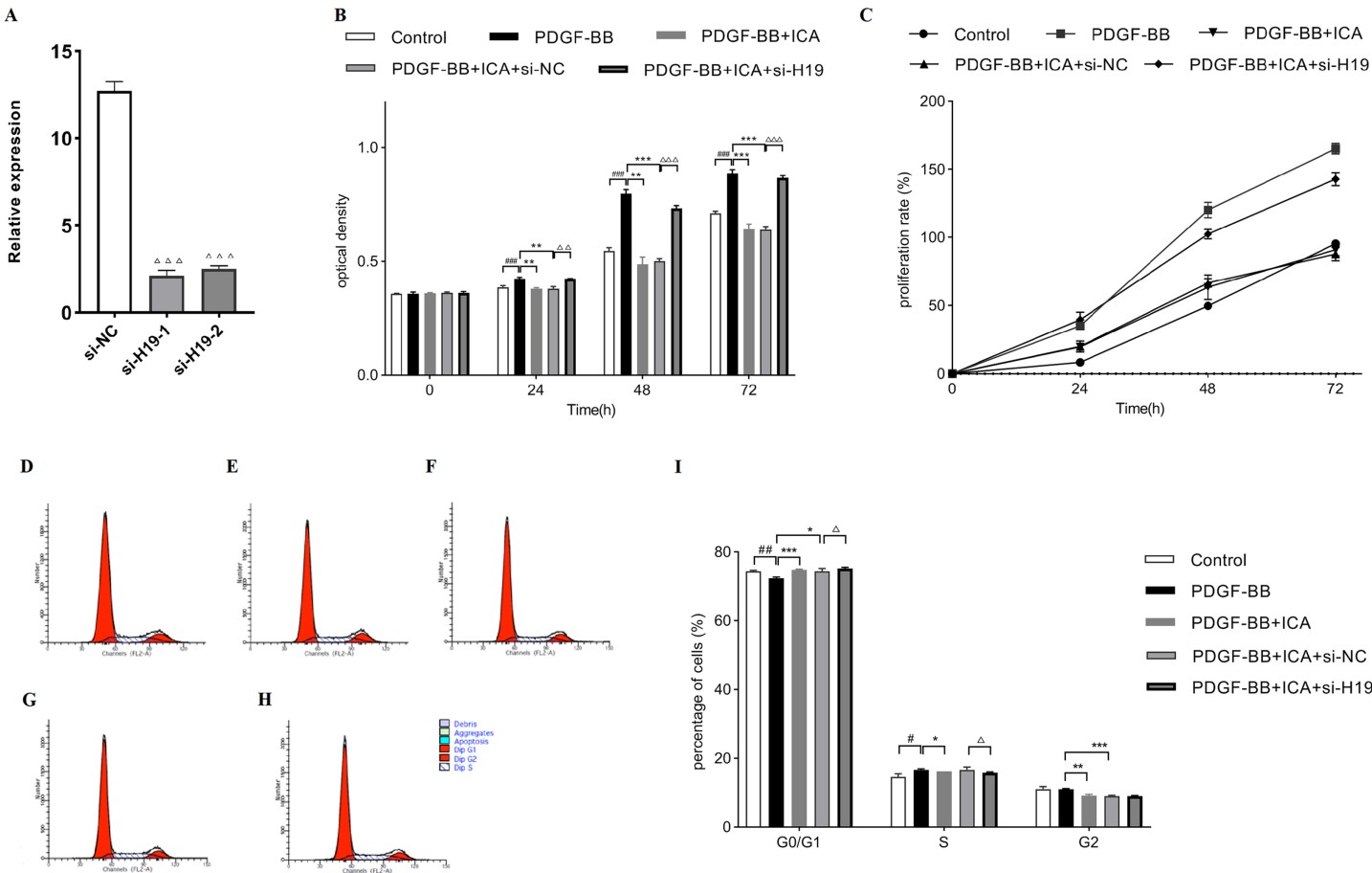

**Figure 5 Down-expression of H19 reversed ICA-induced cell proliferation in PDGF-BB-stimulated RPE cells.** (A) Transfected efficiency of si-H19 or si-NC on H19 expression was detected by qPCR in RPE cells. (B) Cell proliferation of ICA-treated RPE cells with si-H19 or si-NC transfection was measured by MTS assay at 0, 24, 48 and 72 h. Bar graphs show the OD of all groups. (C) The cellular proliferating ability was assessed by the proliferation curve. (D–H) The distribution of cell cycle in RPE cells with si-H19 or si-NC transfection was assessed by flow cytometry; (D) Control group; (E) PDGF-BB group; (F) PDGF-BB + ICA group; (G) PDGF-BB + ICA + si-NC group; (H) PDGF-BB + ICA + si-H19 group. (I) Bar graphs show the percentage of cells at each stage ($n = 3$). OD, optical density. Control: blank control group without PDGF-BB; PDGF-BB: 20 ng/ml PDGF-BB; PDGF-BB + ICA: 20 ng/ml PDGF-BB + 40 μM ICA; PDGF-BB + ICA + si-NC: 20 ng/ml PDGF-BB + 40 μM ICA + si-NC; PDGF-BB + ICA + si-H19: 20 ng/ml PDGF-BB + 40 μM ICA + si-H19. OD, optical density. Data are expressed as mean ± SD. [#]$P < 0.05$, [##]$P < 0.01$, [###]$P < 0.001$ vs. control group; [*]$P < 0.05$, [**]$P < 0.01$, [***]$P < 0.001$ vs. PDGF-BB group; [Δ]$P < 0.05$, [ΔΔ]$P < 0.01$, [ΔΔΔ]$P < 0.001$ vs. PDGF-BB + ICA + si-NC group.

percentage of cell cycle distribution after ICA treatment was significantly reduced by H19 knockdown (Figs. 5D and 5E).

## ICA affected expression levels of cell cycle regulators in PDGF-BB-stimulated RPE cells via enhancing the expression of H19

Finally, expressions of cell cycle-related factors were determined to clarify the suppressive effects of ICA on cell proliferation via enhancing the expression level of H19. As shown in Fig. 6A, H19 knockdown decreased the mRNA levels of p53 and p21 in ICA-treated RPE cells, meanwhile, H19 knockdown reversed the reducing effects of ICA on mRNA levels of CDK4, CDK6, and cyclin D1. Consistently, si-H19 transfection dramatically reversed the regulatory effect of ICA on p53, p21 CDK4, CDK6 and cyclin D1 expression

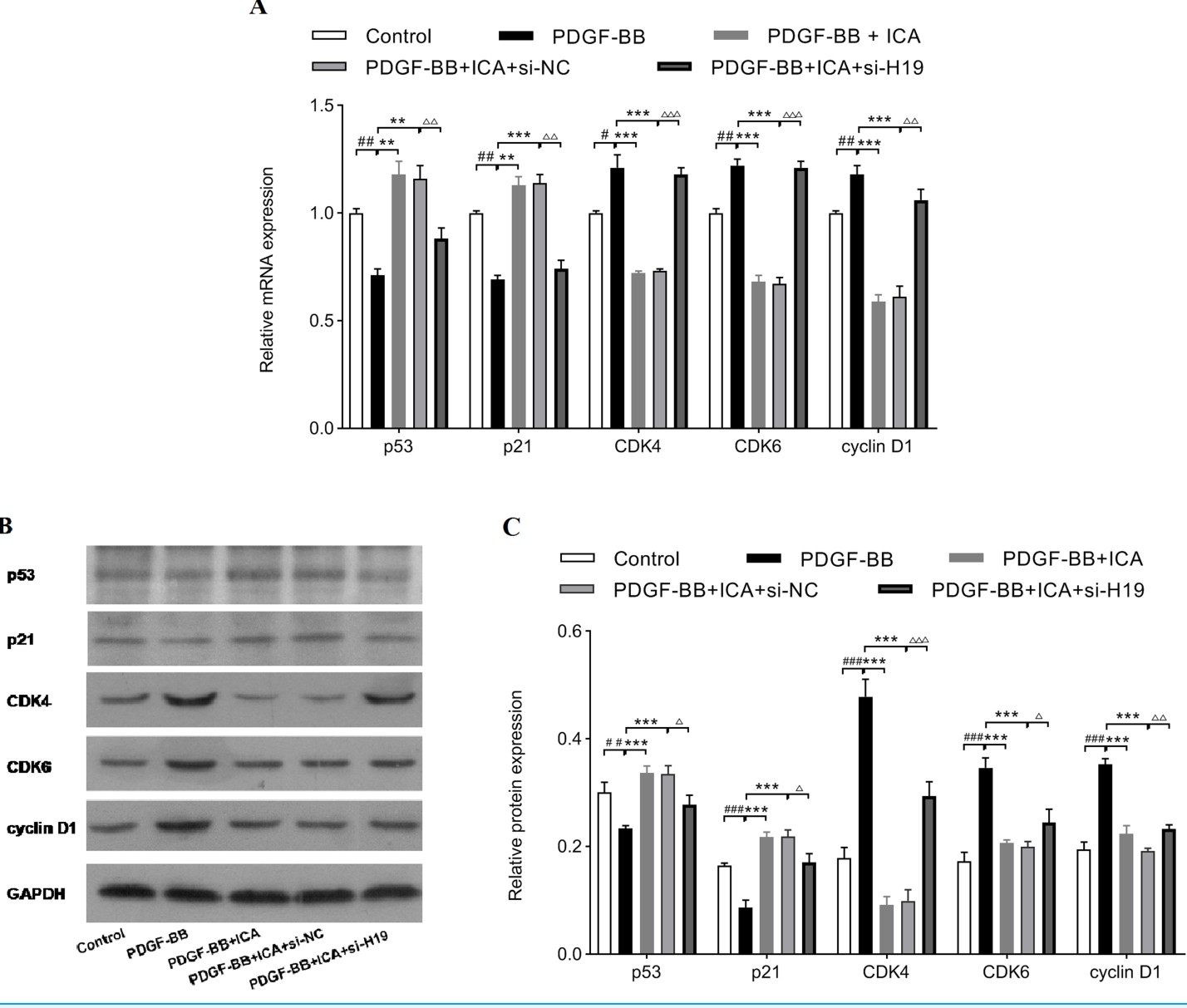

**Figure 6** **Down-expression of H19 reversed ICA-induced changes in cell cycle-related factors expression.** The mRNA levels determined by qPCR (A), and the protein levels detected by western blot assay (B and C). Data are expressed as mean ± SD ($n = 3$). $^{#}P < 0.05$, $^{##}P < 0.01$, $^{###}P < 0.001$ vs. control group; $^{**}P < 0.01$, $^{***}P < 0.001$ vs. PDGF-BB group; $^{\Delta}P < 0.05$, $^{\Delta\Delta}P < 0.01$, $^{\Delta\Delta\Delta}P < 0.001$ vs. PDGF-BB + ICA + si-NC group.

at protein levels (Figs. 6B and 6C). Collectively, these results strongly supported that ICA could modulate the expressions of cell cycle-associated proteins by increasing the expression level of H19 in PDGF-BB-stimulated RPE cells.

## DISCUSSION

Retinal pigment epithelium cells play diverse critical roles in visual function and eye development, they are highly specialized monolayer cells and form the blood-retinal

barrier between the light-sensitive outer segments of the photoreceptors and choroidal vasculature. Excessive proliferation of RPE cells is triggered by specific inflammatory cytokines and growth factors. PDGF has been implicated in vascular proliferative retinopathies, such as DR, and in nonvascular retinopathies, such as PVR (*Cui et al., 2009*). Moreover, cumulative studies have shown that PDGF-BB is a major mitogen for RPE cell proliferation and directly resulted in the development of PVR. *Mori et al. (2002)* generated hemizygous rhodopsin promoter/PDGF-B (rho/PDGF-B) transgenic mice with retina-specific expression of PDGF-B, high retinal expression of PDGF-B results in elevated proliferation of RPE cells and traction retinal detachment. Additionally, *Akiyama et al. (2006)* observed that intravitreous injections of anti-PDGF-B aptamers significantly suppressed retinal detachment and ERMs formation in rho/PDGF-B transgenic mice. Several intracellular signal cascades activated by PDGF-BB have been verified to play key roles in regulating of RPE cell proliferation. The PI3K/Akt and MAPK pathways have been shown to be involved in the mechanisms of various cells proliferation, of which RPE cells are regarded as the predominant target (*Zhang et al., 2019*; *Li et al., 2018b*). *Zhang et al. (2019)* reported that PDGF-BB induces RPE proliferation via PDGF receptor and its downstream component PI3K/Akt pathway, while ERK1/2, p38 and JNK signals are modulated to control the progression of cell cycle. In line with the findings, *Li et al. (2018b)* demonstrated that miR-27b promoted RPE cell proliferation by inhibiting PI3K/Akt/mTOR pathway. Hence, we chose to conduct the recent experiment with PDGF-BB, providing a simulated pathologic condition for RPE cells.

Recent efforts have been directed toward the biochemical inhibition of cellular proliferation in PVR, but the effect is not ideal. ICA is a pharmacologically active flavonoid that can suppress growth and promote apoptosis in a variety of malignant cells (*Sun et al., 2016b*; *Ren, Zhu & Liu, 2018*). Additionally, ICA prevents against mitogen-induced aberrant proliferation in other cell lines. A prior study found that ICA treatment evidently ameliorated angiotensin II (AngII)-induced cell proliferation in human brain vascular SMCs (HB-VSMCs) (*Dong et al., 2019*). Meanwhile, our colleagues demonstrated that oxidized low density lipoprotein (ox-LDL) significantly increases cell viability of HA-VSMCs, yet ICA suppressed ox-LDL-induced HA-VSMC proliferation in a concentration-dependent manner (*Hu et al., 2016*). Previous studies have clearly demonstrated that ox-LDL can promote cell proliferation via activating PDGF receptor-$\beta$ (PDGFR-$\beta$) pathway (*Vindis et al., 2006*). These evidences suggested that ICA affects cell proliferation probably through the inhibition of PDGFR-$\beta$ activation. ICA has been widely used to treat various diseases such as cardiovascular disease, osteoporosis and rheumatoid arthritis (*Li et al., 2015*; *Zhang et al., 2015*; *Fang & Zhang, 2017*), while it was reported that ICA attenuates streptozotocin-induced DR conditions depending upon concentrations both in vitro and in vivo (*Xin et al., 2012*). Comparatively, whether ICA could affect cell cycle progression and proliferation in RPE cells remain unknown. Therefore, it gives us interest to explore the effect of ICA on PDGF-BB-stimulated RPE cell behaviors.

LncRNAs arise more attention of ophthalmologists since they contribute to the regulation of RPE cells in various aspects (*Kutty et al., 2018*; *Ye, Li & He, 2017*; *Chen et al., 2017*). To date, several lncRNAs have been identified to play important roles in regulating cellular proliferation and apoptosis in RPE cells (*Zhou et al., 2015*, *2018*; *Li et al., 2018d*; *Yang et al., 2016*), such as lncRNA BDNF-AS, nuclear factor-κB (NF-κB) Interacting LncRNA and Metastasis Associated Lung Adenocarcinoma Transcript 1 (MALAT1). MALAT1 is one of the best-defined lncRNAs with many functions. *Yang et al. (2016)* confirmed that knockdown of MALAT1 significantly hampered the proliferation of RPE cells after treated with transforming growth factorβ1. Likewise, *Zhou et al. (2015)* reported the same role of MALAT1 in regulating RPE cell proliferation upon the stimulus of tumor necrosis factor α (TNF-α). MALAT1 was identified as a PVR-related lncRNA through microarray analysis with clinical ERMs samples, the elevated MALAT1 levels in peripheral blood of PVR patients were verified also. These parallel results indicated that MALAT1 might be a potential target for the diagnosis and therapy of PVR diseases.

As one of earliest identified lncRNAs, H19 is a key regulator of proliferation in multiple cells. However, the expression levels and regulatory effects on cell proliferation of H19 are discordant, even in the same cell type. For instance, Li and colleagues demonstrated that H19 exerted proapoptotic and anti-proliferative effects in HA-VSMCs leading to subsequent abdominal aortic aneurysm development. The up-regulated expression profile of H19 could also be observed in human aneurysm tissue samples. Since it has been well established that transcription factor p53 reduces cell proliferation in response to stress, molecular mechanism experiments conducted by *Li et al. (2018a)* revealed that H19 could prevent murine double minute 2-mediated reduction of p53 via interacting with hypoxia-induced factor 1α, which in the hypoxic environment of aneurysm triggers apoptosis in HA-VSMCs. Paradoxically, *Zhang et al. (2018a)* stated that H19 depletion could induce apoptosis and suppress proliferation in ox-LDL-induced HA-VSMCs. Thus, it is imperative to better understand the mechanism of H19-mediated cellular proliferation.

H19 is also highly expressed in diverse ocular cell lines. *Klein et al. (2016)* explored the role for cofactors of LIM domain proteins (CLIM) in regulating the proliferative potential of corneal epithelial progenitors and identified CLIM downstream target H19 as a negative regulator of corneal epithelial proliferation. A recent study showed the regulatory effect of H19 on endothelial-mesenchymal transition (EndMT) in proliferative DR. It manifested that the expression of H19 was down-regulated either in high glucose-induced retinal endothelial cells or in diabetic mice; over-expression of H19 reversed high glucose-induced EndMT markers both in vitro and in vivo. Moreover, the expression level of H19 is significantly lower in the diabetic vitreous humor vs healthy control (*Thomas et al., 2019*). Alteration of p21 expression has been reported to be an independent indicator of cell proliferation in tumorigenesis (*Saiz-Ladera et al., 2014*). *Zhang et al. (2018b)* investigated the modulating function of H19 in retinoblastoma, enhanced expression of H19 could induce retinoblastoma cell cycle arrest and promote cell proliferation. Their results also demonstrated that H19 could upregulate p21 expression via competitively binding to miR-17-92 cluster.

As previously reported in the above ocular cell types (*Klein et al., 2016*; *Thomas et al., 2019*; *Zamir-Nasta et al., 2018*), we observed similar negative effect of H19 on RPE cell proliferation. Specifically, our present study found that the level of H19 was reduced upon the stimulus of PDGF-BB in RPE cells whereas H19 expression was significantly increased with ICA treatment. Thus, we further explored the role of H19 in RPE cells by knocking down H19 using siRNA. Functional experiments revealed that H19 deletion obviously reversed the repressive action of ICA on cell cycle progression and proliferation, indicated that H19 might play an essential role in the anti-proliferative activity of ICA. As stated above, H19 inhibited cellular proliferation by regulating the expression levels of p53 or p21 (*Li et al., 2018a*; *Zhang et al., 2018b*). Of note, our data confirmed the regulation of H19 acting on p53 and p21 expression in RPE cells. As one of the p53 downstream genes, p21 inhibits the activity of cyclin/CDK complex, which in turn results in cell cycle arrest. In fact, CDK4/6 plays a necessary role in the G1-S transition by associating with cyclin D and phosphorylating the retinoblastoma protein (*Saiz-Ladera et al., 2014*; *Zamir-Nasta et al., 2018*). Our investigations revealed that ICA could repress the expression of CDK4, CDK6 and cyclin D1, thereby inducing G1-S arrest directly in PDGF-BB-stimulated RPE cells. H19 was responsible for the observed inhibitory effect of ICA, providing a rationale for the treatment of PVR.

Despite these promising results in recent research, the understanding of H19-mediated cellular proliferation in depth is inadequate. The underling mechanisms involved in the regulatory function of H19 are complex, different roles have been described for H19 in regulating various stages of gene expression, mostly associated with its behavior as the precursor of miR-675 (*Liu et al., 2016*). Besides the H19/miR-675 signaling axis to modulate target genes, H19 can function as competing endogenous RNAs (ceRNAs) by base-paring to and sequestering definite miRNAs (*Huang et al., 2017*), whereas H19 can alter mRNAs stability by interacting with specific RNA binding proteins (RBPs) (*Giovarelli et al., 2014*). It will be interesting to explore whether H19 can interact with partners such as miRNAs or RBPs to regulate the expression of cell proliferation-related genes in RPE cells. Further studies, which take the detailed molecular relationships into account, will need to be undertaken. Another limitation is that this present study clarified the effect of ICA on cell growth only by accessing proliferative ability of RPE cells, but the analysis on cell apoptosis was not conducted. Further verifications and reasonable analyses are still worthwhile to explore how ICA modulates RPE cell apoptosis, and the role of H19 in the ICA-mediated effect.

## CONCLUSION

In summary, our work contributed to elucidating the potential target of ICA in RPE cell proliferation, hinting the probable application of ICA in preventing PVR. The finding suggested that H19 participated in the proliferation of RPE cells. ICA administration induced a G0/G1 arrest and hampered cell proliferation in PDGF-BB-stimulated RPE cells, and this may depend on its positive regulation of H19 expression. Nonetheless, the exact regulatory mechanisms of H19 in the context of RPE cell proliferation are still indispensable to be discussed in future.

### Funding

This work was supported by the Youth Development Fund of First Hospital of Jilin University (JDYY10201945). The funders had no role in study design, data collection and analysis, decision to publish, or preparation of the manuscript.

### Grant Disclosures

The following grant information was disclosed by the authors:
Youth Development Fund of First Hospital of Jilin University: JDYY10201945.

### Competing Interests

The authors declare that they have no competing interests.

### Author Contributions

- Yibing Zhang conceived and designed the experiments, performed the experiments, analyzed the data, authored or reviewed drafts of the paper, and approved the final draft.
- Min Li performed the experiments, prepared figures and/or tables, and approved the final draft.
- Xue Han analyzed the data, prepared figures and/or tables, and approved the final draft.

### Data Availability

The raw data is available in the Supplemental Files.

### Supplemental Information

Supplemental information for this article can be found online at http://dx.doi.org/10.7717/peerj.8830#supplemental-information.

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
