# Peer review of "Icariin affects cell cycle progression and proliferation of human retinal pigment epithelial cells via enhancing expression of H19"

_PeerJ, doi:10.7717/peerj.8830_

## Round 0.1 · original submission · Major Revisions

The manuscript is well written, the subject and experimental design are suitable to Peer J. Your findings point icariin as a putative candidate to PVR treatment if further studies confirm its efficacy on in vivo models. Please, answer the queries of reviewers. Pay special attention to statistical issue with cell cycle experiment.

Reviewer 1 ·

Basic reporting

In our opinion, language is acceptable, and the manuscript properly refers to the literature. We request some modifications, as follows:

Please, avoid using the terminology “dose” while working with cells.

Figure 1. The authors claim a concentration-dependent antiproliferative effect of ICA. However, there is no evidence of a statistical difference between each concentration used. Also, Fig 1 A and B must be cited in the results section, though they refer to the same description in the text.

Figure 4 legend merits revision. It neither matches the figure nor the results described on page 11, lines 160-164.

Experimental design

This study falls within the scope of the journal, and the methods are described in sufficient detail. Critical modifications to improve the quality of this study and to increase the potential readership by the journal audience are as follows:

The study does not describe the kind of cell death ICA induces. In our opinion, the antiproliferative assay must be associated with further experiments to indicate the cell death pattern. (for instance, please see DOI:10.1016/j.ejphar.2018.11.001).

What morphological alterations does ICA induce? Does H19 reverse them?

Validity of the findings

There is no citation/description of Figure 2 in the text. Regarding the same figure, the authors indicate that ICA suppressed the cell cycle progression. However, there is no evidence of such a result. Cells on G0/G1, S or G2 phases seem to be quantitatively the same despite the treatment. Are RPE cells susceptible to ICA? What is the duplication time of those cells? We believe the results do not support the conclusions concerning the cell cycle. An example of an altered cell cycle is found in DOI: 10.2174/1566524017666171106115655.

Additional comments

This study investigated the inhibitory effects of the flavonoid icariin (ICA) on retinal pigment epithelial cells against platelet-derived growth factor (PDGF)-BB-induced cell proliferation. Additionally, it explores the influence of H19 on ICA-induced antiproliferative effect. In our opinion, the manuscript is descriptive. How ICA induces cell death pattern merits investigation. We point out some concerns.

Reviewer 2 ·

Basic reporting

- The article was written in clear, unambiguous, technically correct text.
- Literature references was appropriately referenced.
- The structure of the article was conform to the journal acceptable format.
- The results were relevant and clarify the hypothesis, but some terms have been possibly used inappropriately.

Experimental design

- The methods were described with sufficient detail and information to replicate and were appropriately chosen to respond to the research hypotheses.

Validity of the findings

No comment

Additional comments

1. Interestingly, the IC50 values must be shown in Fig. 1.
2. Was the MTS assay performed to evaluate cytotoxicity against ARPE-9 without estimulation of PDGF-BB? What is the IC50 value?
3. The cytotoxicity of ICA should be evaluated against other cell lines in order to evaluate its selectivity.
4. It is interesting to clarify if ICA induced a cytotoxic or cytostatic effect against ARPE-9 cell line. MTS is not the most appropriate method to clarify this. I suggest that you use SRB assay if possible. It is important to evaluate a possible cell cycle arrest relationship with induction of cell death. It would be interesting?
5. In addition, the statistics need to be added to cell cycle data in Fig. 2 and Fig. 5. Could the raw data be sent?
6. In the Line 158 – ICA increased or decreased CDK4, CDK6 and Cyclin D1 protein levels? In Fig. 3 B-C a decrease in CDK4, CDK6 and Cyclin D1 protein levels by ICA was observed.
7. The effect of ICA on PDGF-BB-stimulated RPE cells always returns to basal levels of unstimulated RPE cells. Would it be appropriate to use the terms "upregulate" or "down-regulate" when it comes to the effects of ICA on PDGF-BB-stimulated RPE cells?

Reviewer 3 ·

Basic reporting

1. The work presented is important and well written, however with a few textual inconsistencies and some minor errors in English, which should be thoroughly reviewed.
2. Throughout the text the term dose could be replaced by concentration.

Experimental design

1. I would like to know about the choice of concentrations used (10, 20 and 40µM).
2. Flow cytometry tests did not show the amount of events acquired.
3. Western blot and PCR tests also did not show the amount of sample loaded.

Validity of the findings

Figures 2 and 5. The authors do not show the values or standard deviation bars, as well as the asterisks that symbolizes statistical differences in cell cycle evaluation.

Additional comments

no comment

---

## Round 0.2 · Minor Revisions

Thank you for your responses to the issues addressed by the reviewers. However I have noticed you did not add the apoptosis information in the manuscript (nor in methods, results, discussion or conclusion). What was the flow cytometry assay performed to evaluate apoptosis? This information is missing. This data reinforce your claim about icariin effects and must be properly included in the manuscript. I recommend this figure be part of the manuscript and not supplementary material.

Reviewer 1 ·

Basic reporting

No further comments.

Experimental design

No further comments.

Validity of the findings

No further comments.

Additional comments

Thank you for the rebuttal to the comments we raised.

Reviewer 3 ·

Basic reporting

no comment

Experimental design

no comment

Validity of the findings

no comment

Additional comments

The authors answered all the questions satisfactorily

---

## Round 0.3 · accepted · Accept

I'm glad to inform your manuscript is accepted for publication in PeerJ. Your review and answers to issues pointed out through the review process were fully satisfied.